# What Do Environmental Flows Mean for Long-Term Freshwater Ecosystems' Protection? Assessment of the Mexican Water Reserves for the Environment Program

**Sergio A. Salinas-Rodríguez** [1,2,*], **Everardo Barba-Macías** [3], **Dulce Infante Mata** [4], **Mariana Zareth Nava-López** [2], **Iris Neri-Flores** [5], **Ricardo Domínguez Varela** [2] **and Ignacio D. González Mora** [2]

1 Water Management Department, Civil Engineering and Geosciences Faculty, Delft University of Technology, 2628 CN Delft, The Netherlands
2 WWF-Mexico, CP 06100 Mexico City, Mexico; mnava@wwfmex.org (M.Z.N.-L.); rdominguez@wwfmex.org (R.D.V.); igonzalez@wwfmex.org (I.D.G.M.)
3 Sustainability Sciences Department, El Colegio de la Frontera Sur, Unidad Villahermosa, CP 86280 Villahermosa, Mexico; ebarba@ecosur.mx
4 Sustainability Sciences Department, El Colegio de la Frontera Sur, Unidad Tapachula, CP 30700 Tapachula, Mexico; dinfante@ecosur.mx
5 Escuela Nacional de Estudios Superiores, Unidad Merida UNAM, CP 97357 Yucatan, Mexico; irisneri@gmail.com
* Correspondence: s.a.salinasrodriguez@tudelft.nl; Tel.: +52-55-5286-5631

**Abstract:** Almost a decade ago, the Mexican government targeted to establish environmental water reserves (EWR)—a volume of water allocated for ecological protection based on the Environmental Flow Mexican *Norm* (eflows, NMX-AA-159-SCFI-2012, ratified in 2017)—in strategic low-pressured for water use and high conservation importance river basins throughout the country. To date, 12 EWRs have been declared for up to 50 years, which encompass 295 river basins and ~55% of the national mean annual runoff (MAR). In this article, we conducted a quality evaluation of the EWRs established. First, the EWR level was analyzed against the MAR and according to wider hydrological conditions. The EWR fulfillment was evaluated by comparing the volumes enacted against the theoretical (Norm implementation). Our findings revealed that independently of individual and regional water use and conservation merits context, ~75% of the EWRs met theoretical volumes at least at an acceptable level, of which medians ranged from 24% to 73% MAR (natural parametrization and A–D environmental objectives). These outcomes prove the usefulness and consistency of the Mexican strategic hierarchical approach for eflow assessments. We aim for them to be considered as the baseline for future on-site eflow implementation and environmental water policy assessments, to show the nationwide potential benefits for protecting free-flowing rivers and to encourage a regional escalation of the strategy.

**Keywords:** environmental flows; environmental water reserve; hydrological region; national program; quality assessment; reference values; river basin

## 1. Introduction

The water fluxes throughout the atmosphere, from oceans to continents, have driven physical, ecological, and societal processes; they have sustained life all over the world for millennia. The aquatic ecosystems throughout river basins conduct and store water that has been a source of prosperity in the environment. Despite that the freshwater ecosystems represent ~2% on Earth's surface, by their location in the landscape they possess around 10% of all described animals and one-third of known vertebrate species [1–5]. It is widely accepted that the rivers, lakes, lagoons, and other wetlands provide a wide array of ecosystem services that sustain people's welfare, yet degradation and biodiversity loss on these ecosystems have occurred at alarming rates, far beyond that in comparison with marine

and terrestrial [6,7]. According to World Wildlife Fund's (WWF) Living Planet Report [6,7], ~90% of the surface of the wetlands and ~40% of the abundance of their dependent species have been lost, and only 37% of the long rivers remain free-flowing at a global scale [8]; all this is largely due to habitat degradation, flow modification, or overexploitation including unsustainable water abstraction. To date, an estimated 2.8 million dams have been built, more than 3700 are currently planned or under construction for hydropower generation (>1 MW), and there is 500,000 km of rivers and canals regulated or created for navigation and transport, or to address water-related environmental services demanded by society (i.e., domestic, irrigation, industrial use) [8]. Given the present and future pressure over these ecosystems [6,7], it is urgent to take action to foster stronger policies on protecting the freshwater ecosystems and their biodiversity [4,9,10] and hold the ground for a suitable balance between nature conservation and water infrastructure operation and new developments.

The environmental flow (eflow) science—underlying science of environmental water allocations—has advanced significantly in recent decades [11–13]. Defined as the quantity, timing, and quality of freshwater flows and levels necessary to sustain aquatic ecosystems which, in turn, support human cultures, economies, sustainable livelihoods, and well-being [14], the implementation of eflows has been targeted as a top action towards science-based freshwater ecosystems conservation and management, urgently needed to bend the curve on biodiversity loss [5,10,15,16]. Furthermore, their implementation, together with a complementary suite of policy, legislative, regulatory, financial, scientific, and cultural measures holds the potential for reaching trade-offs among sustainable water usage and ecosystem protection [5,10–16]. The present research article focuses on the assessment of the implementation of the Mexican eflow policy in 2012–2018, which aimed to enact environmental water reserves (EWR) [17–19], an annual-based volume designated to remain in the environment for ecological protection for up to 50 years [20–22].

The Mexican National Water Reserves for the Environment Program (NWRP) was launched in 2012 by the National Water Commission (CONAGUA) jointly with the alliance of WWF-Fundación Gonzalo Río Arronte I.A.P. and supported by the National Commission of Natural Protected Areas (CONANP) [20,21]. Initially, 189 potential water reserves were identified based on their relatively low pressure for water use and their ecological importance at a basin-scale, and they were strategically targeted to build capacities in eflow assessments, demonstrate their benefit to support healthy rivers, and establish a national system based on the flow regime protection [20,21,23]. These potential reserves were adopted by the last federal administration as environmental water goals in the Mexican Programmatic Plans of Environment 2013–2018 [17], Water 2014–2018 [18], and Climate Change 2014–2018 [19].

By 2015, eflows detailed assessments in eight pilot zones were concluded with hydrological and holistic methodologies, one EWR was established, and by 2018 nearly 300 EWRs were enacted in 12 hydrological regions, causing this initiative to be in the spotlight in the eflows implementation arena [6,10,11,20,21,24–28]. Although the policy's outcome surpassed the commitment, the quality of the reserves and the associated reference values have not been examined, nor have their potential contribution as a long-term protection measure of Mexican free-flowing rivers been discussed, and the novelty of the work lies in these aspects. In this research article, we aim to assess the quality level of the EWRs, individually (river basin), at a hydrological region level, and as a nationwide system to provide their reference values and discuss the findings in the light of their implications and limitations. We also aim for such results to be used as a baseline for reporting on further progress in environmental water allocations, and as a nationwide case study in eflow implementation.

## 2. Materials and Methods

EWRs were evaluated based on the Mexican Norm that establishes the procedure for environmental flow determination in hydrological basins, also referred to as Standard or NMX-AA-159-SCFI-2012 [29]. This regulatory instrument is a three-level hierarchical

framework that aims to find a balance between water use and freshwater ecosystem conservation [20,30,31]. It sets the ecological, hydrological, and water management principles for both people and nature—from a public policy perspective—to conduct eflow assessments, from relatively simple and cheap (i.e., "look-up tables" and hydrology-based) to more comprehensive and expensive methodologies (i.e., ecohydrology-, habitat simulation- and holistic-based) [21,22,27].

As the main user of the Mexican eflows Norm, and as a standard procedure of the NWRP, CONAGUA implemented it thoroughly in the EWRs established in 2018 [32–42]. For this manuscript, the reserves' quality assessment was conducted based on such outcomes under open access accountability and transparency spirit aiming to contribute the baseline of the National Water Program 2020–2024 environmental water-related goals [43], for which written consent was given [42].

### 2.1. Method Used, Data Requirements, Supporting Indices, and Statistics

In the following Sections 2.1 and 2.2, the general procedure for both the eflow and the EWR quality assessment is described (Figure 1). The EWRs were assessed based on the theoretical eflow requirements according to the methodology stated in the Mexican Standard's Appendix D (application 2). This ecohydrology-based methodology was selected because it is grounded in the frequency-of-occurrence of eflow components, recognized by the environmental water science and a state-of-the-art practice [11,22,24–27]. Monthly-scale of very dry, dry, average, and wet seasonal ordinary low-flows conditions (Solf) were assessed [21,22]. Likewise, a daily-scale flood regime (Fr) encompassing three peak flows at a magnitude of 1-, 1.5-, and 5-year return period was evaluated [21,22]. The characteristic duration (hours) was also calculated, as this flow attribute is required to integrate eflow needs into annual-based volumes (million cubic meters, $hm^3$) [21,22]. Afterward, the EWRs were obtained according to the natural parametrization of the frequency of occurrence of both eflow components, for the low flows set at a 25% of the time for wet, 50% average, 15% dry, and 10% very dry conditions, respectively; and for the flood regime based on the peak flow events' modeled return periods [22]. Likewise, theoretical EWRs were adjusted to a four-tired environmental objectives class system from "A" to "D" according to the frequency factors of occurrence [22] built based on the eflow components occurrence's natural parametrization [22–31]. Based on the method, class "A" means a "very good" desired state of the flow regime, while "B", "C", and "D" refer to a "good", "moderate", and "deficient" state, respectively, as similarly used in the eflows practice [16,21,44–47].

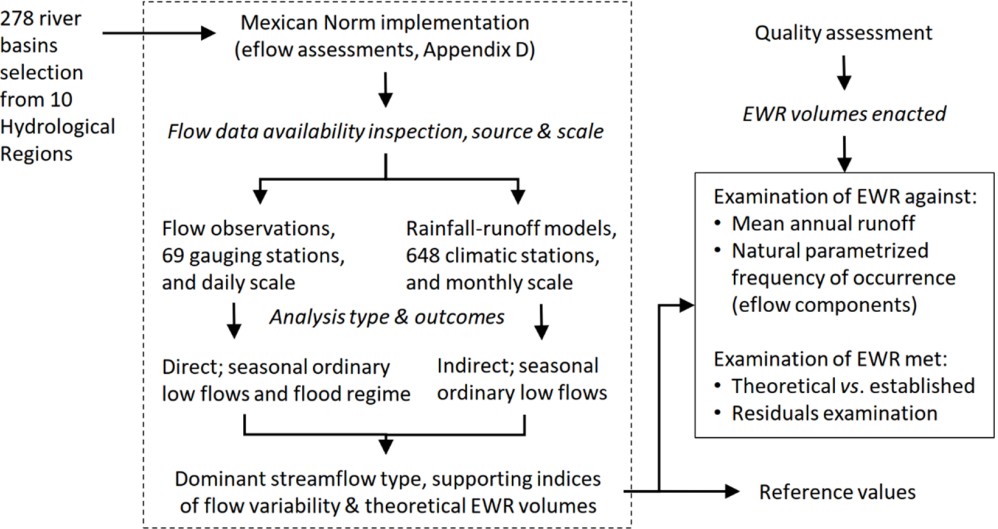

**Figure 1.** General procedure for the environmental flow (eflow) and water reserve assessments.

This novel frequency-of-occurrence-based approach for low flows and flood regime eflow components was also chosen because of its potential for managing freshwater and riparian species exposure to each-time-more intensely extreme conditions, and it contributes to addressing the non-stationarity challenge of the flow regime variability by managing resilience within the limits allowed by the biodiversity [11,22,48–54]. Another reason for the selection of this method was because, among the desktop-based ones provided by the Mexican Norm, this reaches the highest detail of analysis and, therefore, it has been systematically implemented throughout the country; furthermore, consistency of outcomes has been proved when they were examined against a holistic method [21,55–61]. Throughout the article, eflow assessment results from a holistic expert panel for the case of the Usumacinta river are presented to exemplify an evaluation in detail based on the Mexican Standard's Appendix F (Box 1 further in this Section; Boxes 2–4 in Sections 3 and 4).

Eflows from 278 EWR in 10 hydrological regions (HR) were assessed [32–41]: Lerma-Santiago (HR code 12), Río Ameca (14), Costa de Jalisco (15), Costa Grande de Guerrero (19), Costa Chica de Guerrero (20), San Fernando-Soto La Marina (25), Panuco (26), Papaloapan (26), Coatzacoalcos (29), and Grijalva-Usumacinta (30; Figure 2). Although to date 295 EWR have been established, in the present research article, the corresponding reserves from the HR 10 Sinaloa and 11 Presidio-San Pedro were excluded because they were established between 2014–2016. Due to the fact that finding appropriate and suitable flow records throughout the country remains to be challenging, the utilization of the eflows' frequency-of-occurrence approach was split by direct and indirect analyses. Direct analysis refers to where the method was implemented in potential water reserves with available daily flow observed records (69 gauging stations, usable period ranged from 18–60 years; data obtained from CONAGUA's repository ftp://ftp.conagua.gob.mx/Bandas/). In this case, both the low flows and flood regime components were assessed. In contrast, the indirect analysis was conducted in 216 river basins targeted as potential water reserves where flow records had either low quality, reduced length, altered flows, or lack gauging stations. From those, the method was applied for only the low flow component in 211 cases based on monthly-scale rainfall-runoff models built for the water availability studies (648 climatic stations, usable period ranged from 30 to 53 years) [42,62,63]. Although splitting the analysis by direct and indirect implementation limited the scope of the outcomes and brought uncertainty (i.e., flow observations vs. modeled and time resolution), eflow assessment fulfilled the normative requirements.

As in recent research around the topic [21,22,31], complementary indices of flow variability were obtained; these help to the understanding of the regime characteristics as well as annual-based EWR scope and limitations. Streamflow type (flow rate observation or modeled at a unit outlet) was identified due to it is a direct response of the basins to their dominant climates, geography, orographic effects, and EWR's dependency on flow variability [31,44,46]. Dominant streamflow per river basin was labeled according to the following equation applied in daily- and monthly-scale flow duration curves (Q, m$^3$/s):

$$Streamflow\ type \ = \ \begin{cases} ephemeral, & if\ Q > 0.5 \leq 30\% \\ intermittent, & if\ Q > 0.5 > 30\% < 90\% \\ perennial, & if\ Q > 0.5 \geq 90\% \end{cases} \tag{1}$$

In the same line, and as a deeper reference of hydrological variability, additional indices were calculated. These were the coefficient of variation of flows among dry and wet seasons (CV) as an indication of long-term variability, a baseflow index (BFI) representative of short-term variability (ratio of the mean annual baseflow to the mean annual runoff, MAR), and their logical combination for an overall index of variability of flows (CVB = CV/BFI) [21,22,44,46].

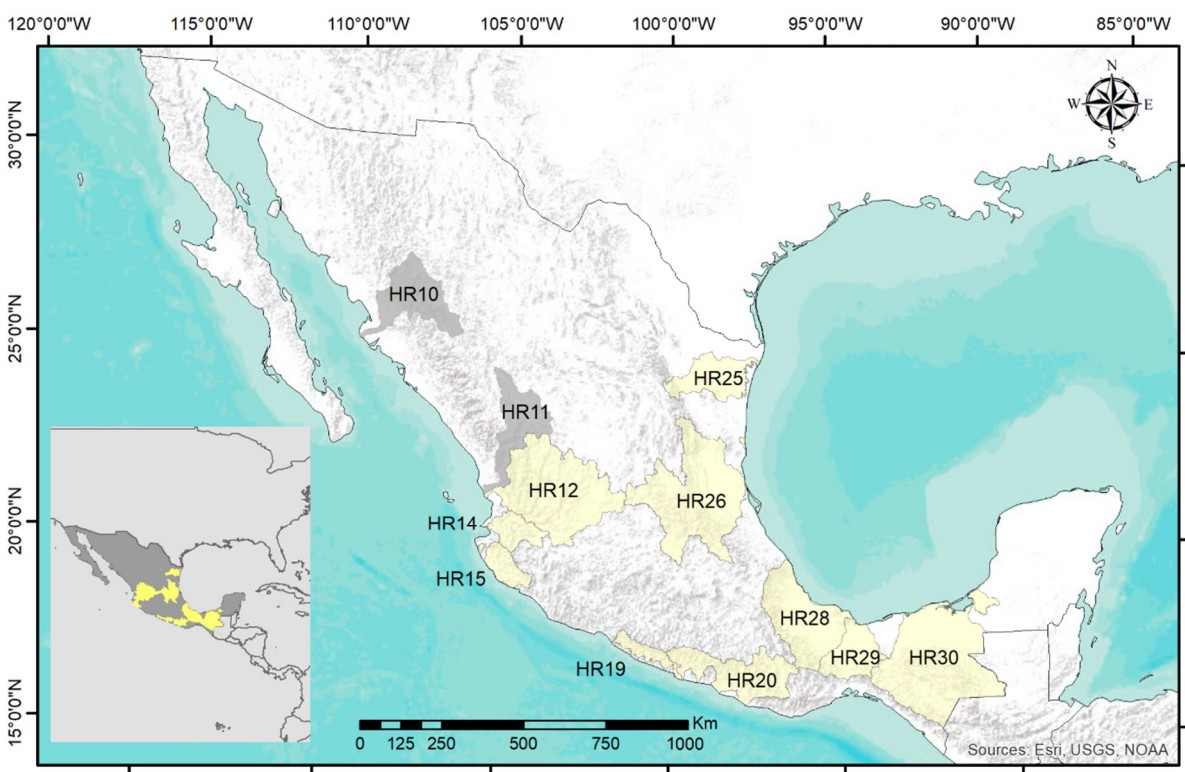

**Figure 2.** Location of the Mexican environmental water reserves established by 2018. Hydrological regions (HR): 10 Sinaloa and 11 Presidio-San Pedro established in 2014–2016, and 12 Lerma-Santiago, 14 Río Ameca, 15 Costa de Jalisco, 19 Costa Grande de Guerrero, 20 Costa Chica de Guerrero, 25 San Fernando-Soto La Marina, 26 Panuco, 28 Papaloapan, 29 Coatza-coalcos, and 30 Grijalva-Usumacinta established in 2018 [32–41].

### 2.2. Nationwide Environmental Water Reserves Quality Assessment and Reference Values

The quality assessment of the EWR established was evaluated in three ways. First, EWR1, the relation of each river basin EWR established volume (hm$^3$) was compared against its MAR. Similarly, EWR2, it was also compared against the corresponding volumes of the low flow conditions' natural-parametrized frequency of occurrence (NatPFoO), and the peak flow events for the case of the method's direct implementation. Both indicators are important, the first because the MAR tends to be the prevailing metric in water management; outcomes in this regard should not exceed 100%. The second indicator is adjusted to a wider set of hydrological conditions and their occurrence, which includes the extremes (from wet-to-very dry low flows to peak flow events from 1- to 5-year return period). In this case, exceeding 100% values mean that the EWR was set with greater volumes than theoretically recommended as a result of the implementation of the eflows Norm.

$$Percentage\ of\ EWR1\ = \frac{EWR\ established}{MAR}\ \times\ 100 \qquad (2)$$

$$Percentage\ of\ EWR2 = \frac{EWR\ established}{NatPFoO}\ \times\ 100 \qquad (3)$$

Second, theoretical-based EWR (NMX-AA-159-SCFI-2012 implementation) per river basin expressed as a percentage of the MAR was compared against the percentage of the annual volume that was officially allocated for environmental use (EWR established). In this case, the indicator means the percentage of the EWR established that met theoretical volumes (EWR met), where 100% represents equality between them, below it means that

the established is lower than the theoretical, and above it means that the established is beyond the recommended. It was calculated based on the following equation:

$$Percentage\ of\ EWR\ met\ =\ \frac{EWR\ established}{EWR\ theoretical}\ \times\ 100 \qquad (4)$$

Third, to complement the previous index, theoretical and established EWRs were subtracted, and the residuals were examined based on ≤5%, ≤10%, and ≤15% levels of difference. Negative values indicate that the EWR established is greater than recommended, therefore this number and proportion of basins was considered an indication of fulfillment.

Finally, reference values of EWRs, as well as CV and BFI, were obtained based on a central range distribution approach (median ± 25% or quantiles 1, 2, and 3). The whole methodological approach was applied in the river basin outlets to ensure consistency with the Mexican regulations and, therefore, the outcomes are provided at a basin-scale [29,32–41,62,63]. A database with all the outcomes described above was developed and uploaded as Supplementary Material to support the present manuscript. The calculations and plots were made in MO Excel and Past 3.0.

**Box 1.** Usumacinta river: Goals, method, and strategic environmental flow arrangements.

> The Usumacinta river basin is a transboundary basin that extends from northwestern Guatemala to southeastern Mexico (Figure 3). It has an area of approximately ~73,000 km$^2$ of which ~31,000 km$^2$ are in the Mexican territory covering the states of Campeche, Chiapas and Tabasco [64]. The Usumacinta, is the most important hydrological basin in the Central American region, as it contains the largest and longest river in all Mesoamerica.
> The Usumacinta receives its name at the junction of the Pasion and Chixoy rivers that descend from the Guatemalan Sierra; downstream is fed by waters of the Lacantun river and delineates the border between Mexico and Guatemala. Overall, the Usumacinta flows through a river network of ~12,800 km from source to mouth, where it intersects with the Grijalva river and drains into the Gulf of Mexico. While most of the upper part of the Usumacinta basin lies in Guatemala (58%), the lower part is exclusively Mexican, implying that Mexico receives the accumulated impacts of the hydrological network and related transformation processes that occur along the river course [65].
> In 2011, in the context of Mexico's NWRP, the Usumacinta basin was identified as a potential water reserve [23] given the basin's low pressure on water resources and the exceptional levels of biodiversity and conservation values. These include many endemic and threatened species, habitat diversity from tropical rainforest and floodplains to extensive wetland areas and large estuarine lagoon systems that depend to a great extent on the river's flow regime. Except for a hydropower plant built in the upper basin (Chixoy river, 390–460 hm$^3$ storage capacity, 275–300 MW effective capacity), the Usumacinta river flow that lies within the Mexican territory remains free from water infrastructure (i.e., connectivity values above 95% of conservation status [8]).
> In 2018, the river's connectivity was protected by establishing an EWR at 90–94% of the mean annual runoff. Eflow determination and characterization of environment conditions, biological cycles, and their relationship with the hydrological regime were studied by implementing Mexican Norm's holistic expert panel approach (Mexican Standard Appendix F). For the study, the Mexican portion of the basin was divided into three zones according to their main hydrological and ecological features. These zones were agreed upon in a multidisciplinary workshop, where in turn, information gaps were identified. The upper region, characterized by a predominance of Cretaceous limestones and sedimentary rocks, was identified as Zone I—Lacantun. The lower-middle region that includes a large alluvial floodplain was identified as Zone II—Jonuta-Catazaja—and the lower region characterized by extensive peatland tropical wetland areas was identified as Zone III—Tres Brazos. For each zone, reference sites were selected where the river's surface and groundwater flows, topography and bathymetry, riparian vegetation, and fish communities were sampled in dry and wet seasons. Additionally, connectivity and habitat availability dynamics in the Ramsar site 1765 Sistema Lagunar Catazajá (*Catazaja Lagoon System*) in Zone II, were studied based on hydraulic modeling and remote-sensing. The results of the eflow study were presented before a multidisciplinary expert panel that was brought to use both their expertise and judgment to develop final eflow recommendations and EWR.

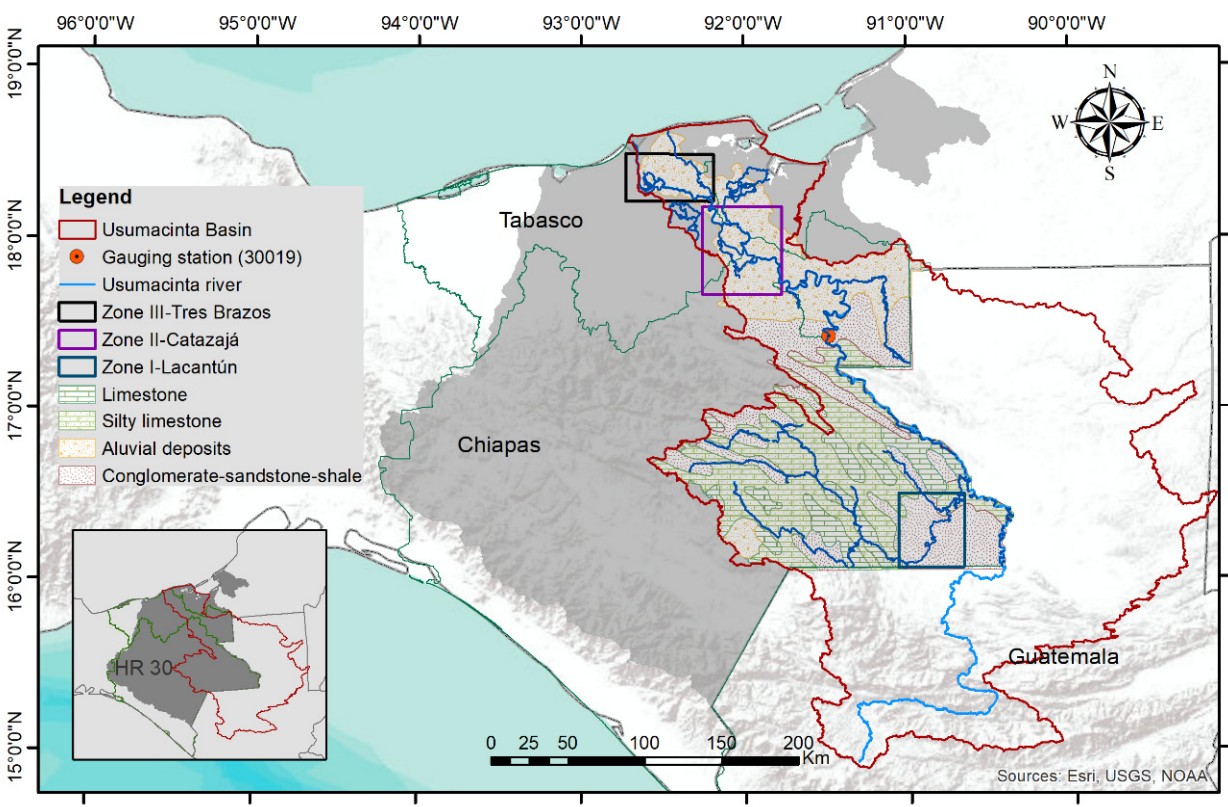

**Figure 3.** Location of the Usumacinta river basin.

## 3. Results

### 3.1. Quality Assessment of the Environmental Water Reserves

In terms of the MAR, the medians of the EWRs established per hydrological region were in Lerma-Santiago 17%, Río Ameca 55%, Costa de Jalisco 54%, Costa Grande de Guerrero 43%, Costa Chica de Guerrero 42%, San Fernando-Soto La Marina 29%, Panuco 15%, Papaloapan 59%, Coatzacoalcos 47%, and Grijalva-Usumacinta 50% (Figure 4). However, the Mexican EWRs were evaluated for different hydrological conditions, and integrated based on their occurrence adjusted to each river basin's environmental objective or management class. A more meaningful metric of comparison is against the natural parametrized frequency of occurrence of such conditions. In this sense, the medians of the EWRs were in Lerma-Santiago 27%, Río Ameca 66%, Costa de Jalisco 72%, Costa Grande de Guerrero 67%, Costa Chica de Guerrero 54%, San Fernando-Soto La Marina 16%, Panuco 23%, Papaloapan 75%, Coatzacoalcos 62%, and Grijalva-Usumacinta 63%. Except for San Fernando-Soto La Marina, in the remaining it is observed that the relative volumes compared to the natural parametrized frequency of occurrence are greater than to the MAR. This observation is a sign that the EWRs have greater meaningfulness to the long-term variability than the MAR that is a less sensitive statistic.

Concerning the number of basins and proportions in which the EWRs established met theoretical volumes, the relative volumes in 169 out of 278 cases (61% of the basins) were ≥90% equality, in 189 (68%) were ≥80%, and in 202 (73%) ≥70% (Figure 5). At a hydrological region level, the medians of percentage EWR met from Lerma-Santiago, Río Ameca, San Fernando-Soto La Marina, Papaloapan, and Grijalva-Usumacinta were calculated at 100% equality, while in Costa de Jalisco, Costa Chica de Guerrero, and Coatzacoalcos regions it was at 88–89%. This means that in general, in these regions the quality of the reserves established is at the same level or very close to the theoretical. In contrast, Panuco showed the lowest level found (55%), while Costa Chica de Guerrero EWRs surpassed the theoretical volumes by almost 50% (149%).

About the examination of the EWRs residuals', in 179 out of 278 basins (64%), the established reserves fulfilled the theoretical at a ≤5% level of difference, while 207 (75%) were ≤10%, and 228 (82%) ≤15% (Figure 6). According to the residuals' medians, the hydrological regions that showed the best performance were Costa Grande de Guerrero, Lerma-Santiago, Río Ameca, San Fernando-Soto La Marina, Papaloapan, and Grijalva-Usumacinta with ≤5%. In a second-middle-level, Costa de Jalisco, Costa Chica de Guerrero, and Coatzacoalcos were identified (6–7%), while Panuco showed the lowest level of performance (13%).

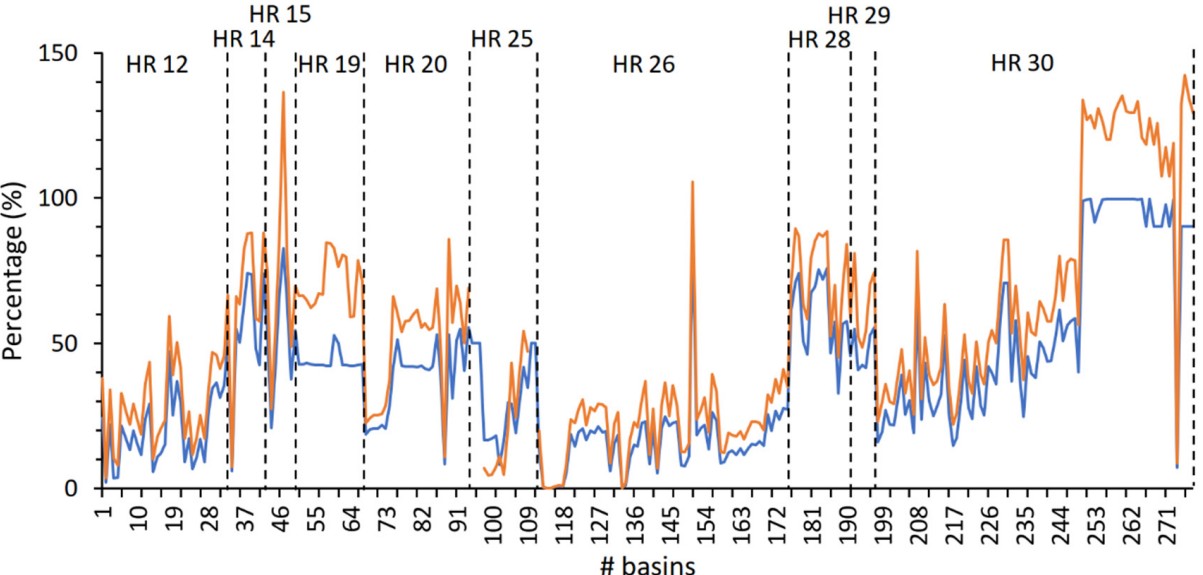

**Figure 4.** Comparative analysis of the environmental water reserves established against the percentage of the mean annual runoff (blue line) and the natural parametrized frequency of occurrence of the low flows and flood regime components (orange line). Hydrological regions (HR), 12 Lerma-Santiago, 14 Río Ameca, 15 Costa de Jalisco, 19 Costa Grande de Guerrero, 20 Costa Chica de Guerrero, 25 San Fernando-Soto La Marina, 26 Panuco, 28 Papaloapan, 29 Coatzacoalcos, and 30 Grijalva-Usumacinta.

*3.2. Environmental Water Reserves' and Flow Variability Indices' Reference Values*

Regarding the EWRs reference values, a central range of 67–78% MAR (median 73%) is observed for the natural parametrized frequency of occurrence, while the characteristic volumes per environmental objectives ranged from 50% to 64% for a class "A" (median 58%), 32–50% class "B" (median 41%), 24–42% class "C" (median 33%), and 15–33% class "D" (median 24%; *n* = 278; Figure 7). On-site environmental objectives are presented in Box 2 as an example of the expert panel evaluation of the Usumacinta river.

Similarly, the flow variability supporting indices ranged from 108% to 202% CV (median 146%) between dry and wet seasons, and 3–16% BFI (median 10%). In Figure 8, their logical combination has shown that the set of EWRs covered the range of cases stated by its theoretical relationship, that is from regions with large baseflow contributions (high BFI) to others subject to droughts that affect both high and low flows (high CV), or to regions better buffered against droughts (low CV) [22,31,44,46]. Furthermore, the overall indicator of flow variability (CVB) from the whole set of EWR ranged from 6% to 75% (median 16%). The reference values from both the EWRs and the flow variability indices differentiated between gauged flow records and the rainfall-runoff models' outcomes are presented in Appendix A. Likewise, the seasonal ordinary low flows and flood regime's reference values in Appendix B.

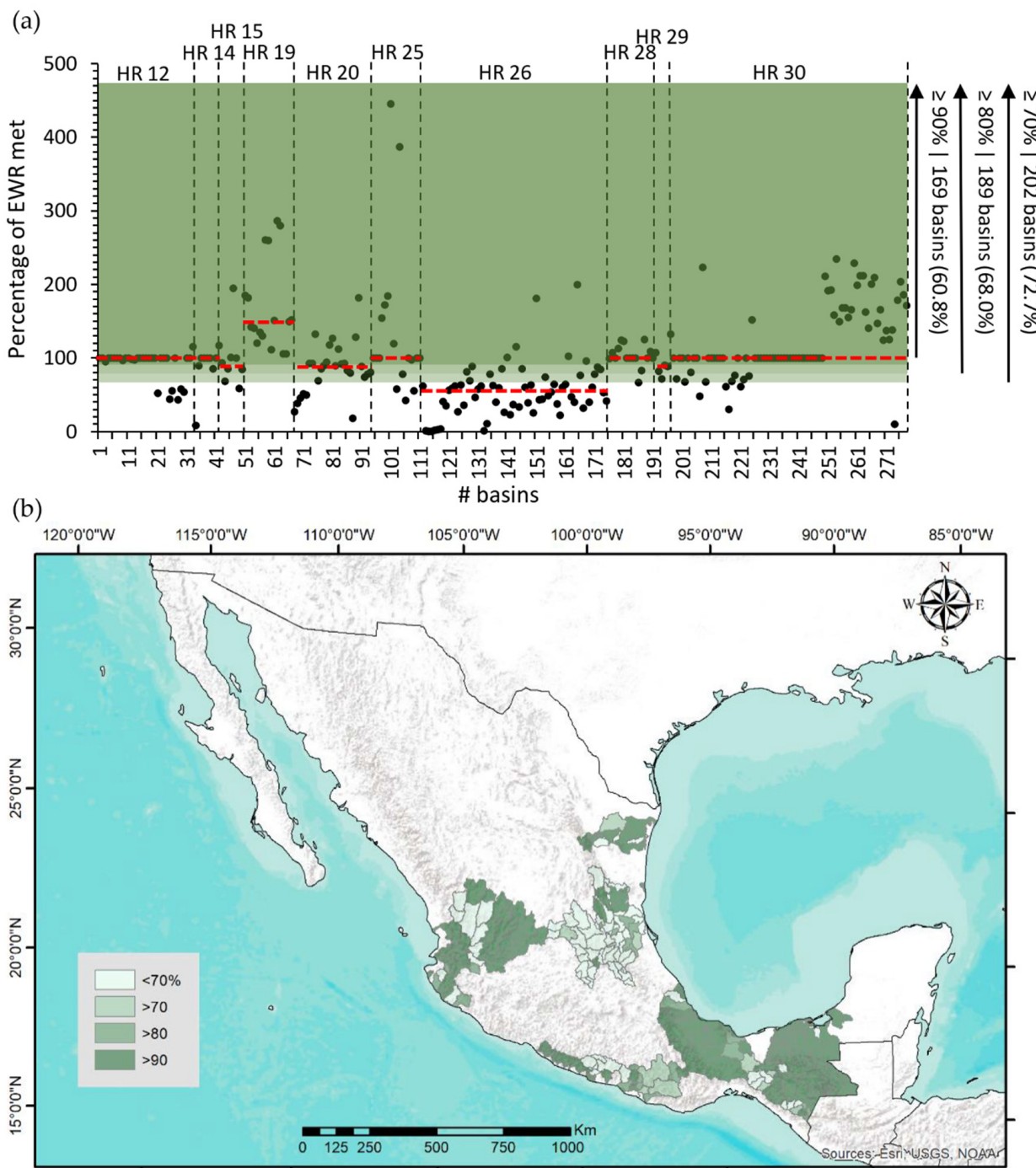

**Figure 5.** Percentage of environmental water reserves (EWR) that met theoretical needs according to the environmental flows' Mexican Norm (NMX-AA-159-SCFI-2012) implementation outcomes. (**a**) Red-dashed horizontal lines represent the median per hydrological region (HR), 12 Lerma-Santiago, 14 Río Ameca, 15 Costa de Jalisco, 19 Costa Grande de Guerrero, 20 Costa Chica de Guerrero, 25 San Fernando-Soto La Marina, 26 Panuco, 28 Papaloapan, 29 Coatzacoalcos, and 30 Grijalva-Usumacinta. Shaded dark green is ≥90% of EWR met, shaded light is ≥80%, and lighter green is ≥70%. (**b**) Geographical representation.

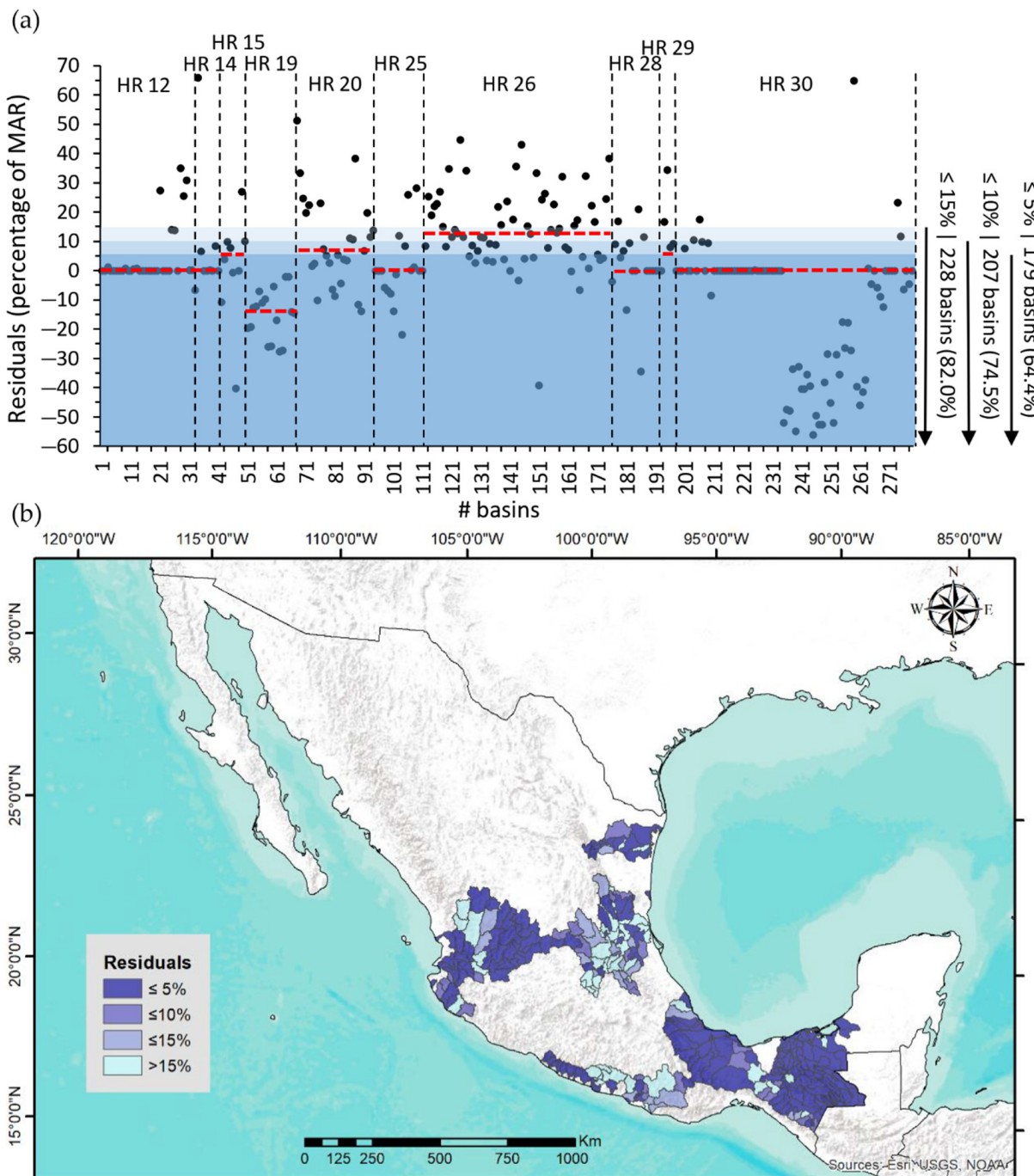

**Figure 6.** Residuals of the environmental water theoretical needs according to the environmental flows' Mexican Norm (NMX-AA-159-SCFI-2012) implementation outcomes against the established volumes (percentage of mean annual runoff, MAR). (**a**) Red-dashed horizontal lines represent the median per hydrological regions (HR), 12 Lerma-Santiago, 14 Río Ameca, 15 Costa de Jalisco, 19 Costa Grande de Guerrero, 20 Costa Chica de Guerrero, 25 San Fernando-Soto La Marina, 26 Panuco, 28 Papaloapan, 29 Coatzacoalcos, and 30 Grijalva-Usumacinta. Shaded dark blue is $\pm$ 5% of difference, shaded light is $\pm$ 10%, and lighter blue is $\pm$ 15%. (**b**) Geographical representation.

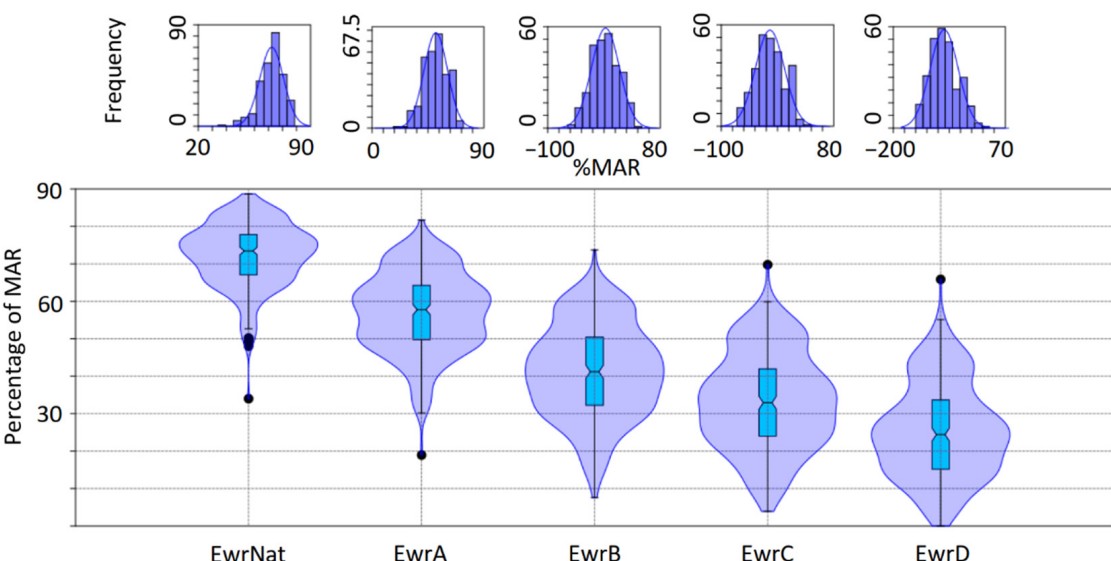

**Figure 7.** Environmental water reserves (EWR) reference values in percentage of mean annual runoff (MAR) based on a central range distribution approach on the environmental flows' Mexican Norm (NMX-AA-159-SCFI-2012) implementation outcomes (*n* = 278). Values are given for the natural parametrized reference (EwrNat) and the environmental objectives (A–D).

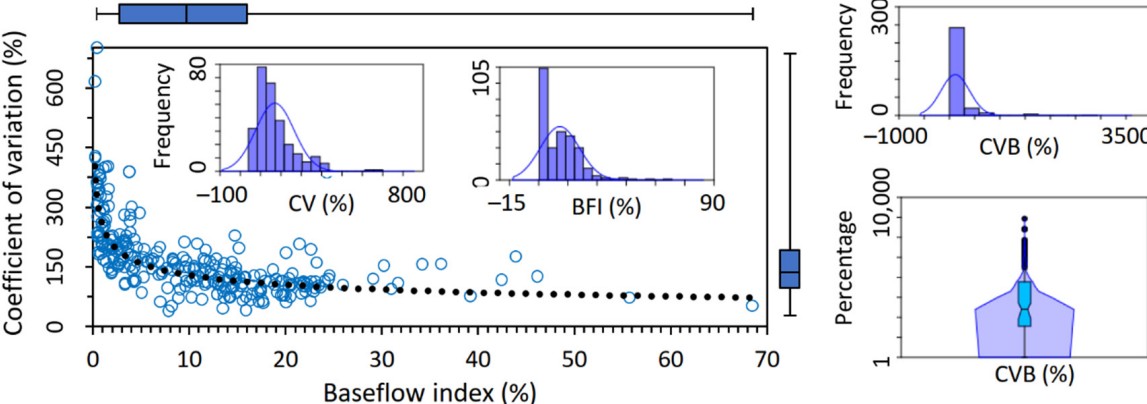

**Figure 8.** Relationship between the coefficient of variation seasons' indicator (CV) and the baseflow index (BFI) ($y = 254.51x^{-0.295}$, $R^2 = 0.56$), the overall index of flow variability (CVB; boxplot displayed at log scale), and their reference values calculated based on a central range distribution approach (*n* = 273).

**Box 2.** Ecological importance, conservation objectives, pressure on water resources, and environmental objectives in the Usumacinta river basin.

The Usumacinta river concentrates high levels of biodiversity and habitats, a large number of ecosystem services, important water resources, and a wide cultural diversity. It harbors more than 20,000 species leading the country's list in terms of vascular plants, freshwater fish, amphibians and birds, and it is second place in reptile species [64]. Specifically for fish, there are 115 species registered, 36% of those are endemic with a high diversity of cichlid and poecilid species [66]. It is because of this biodiversity, that the Usumacinta basin has about 69% of its surface designated for conservation, with 18 federal and state Natural Protected Areas, in addition to two presidential decrees to conserve the rainforest of the region [67] and protect it against hydrocarbons extractions [68]. Likewise, there are Ramsar sites, represented by nine wetlands located mainly in the lower part of the basin. The relevance of establishing EWRs in the Usumacinta basin is largely supported by its exceptional biodiversity and conservation values and the fact that the basin has very low pressure for water use, as only 171.8 hm$^3$ is allocated for consumptive uses that correspond to only 0.29% of the mean annual runoff [29,63]. Final eflow recommendations per reference site were determined in a facilitated workshop and based on the discussion of the current ecological status, ecological importance and sensitivity, conservation objectives, and flow scenarios [65]. The expert panel recommendations are summarized as follows:

**Box 2.** *Cont.*

---

**Reference Site Lacantun**
Ecological importance and conservation objectives: Very High.

- Vegetation: Tropical swamps trees *Inga vera*, *Ceiba* sp., *Pithecellobium lanceolatum*, and *Brosimum ali-castrum*. Palms *Roystonea dunlapiana* and *Attalea butyracea*.
- Fauna: Crocodile (*Crocodylus moreletii*), sea otter (*Lontra longicaudis*), common snook (*Centropomus un-decimalis*), tarpon (*Megalops atlanticus*), Chiapas catfish (*Lacantunia enigmatica*), central-american river turtle (*Dermatemys mawii*), and pale catfish (*Rhamdia guatemalensis*).
- Processes: Species migration, sediment and nutrient transport, longitudinal connectivity through migratory species like the common snook and tarpon as top predators in the system and for regulating the aquatic community, vertical connectivity to guarantee groundwater recharge, and lateral connectivity between the river main stem and permanent and temporary wetlands.

Rationale: It is considered the main groundwater recharge area of the entire Usumacinta basin. The rainforest stores a very significant percentage of Mexico's biodiversity. Of the 9000 species of vascular plants registered for the State of Chiapas, 5000 are found in the Lacandon rainforest. This reference site is the last refuge habitat in Mexico for the scarlet macaw (*Ara macao*), whose feeding habits rely on tree species that are dependent on the flow regime. There are two high ecological important conservation areas: Montes Azules and Lacantun Biosphere Reserves.

Expert recommendation: Setting an EWR is consistent with the current existing conservation instruments and ecological importance. Therefore, experts recommended the conservation of ordinary low flows at 99.5% of the mean monthly flow integrity during both dry and wet seasons (January to May–June to December). It was also recommended that the conservation be 100% of the flood regime integrity to allow the occurrence of peak events, thus promoting seed dispersal processes of flow-dependent tropical swamp trees *Inga vera* and *Pithecellobium lanceolatum*, and limiting the dispersal of opportunistic species with invasive potential that usually appear during long drought periods. The overflowing of the river maintains the lateral connectivity between permanent and temporary wetlands, allowing sediment transport and sustaining biodiversity, in addition to contributing to the creation of microhabitats for the refuge and growing of fish. Migratory movements are a source of important exchange of nutrients and energy transference maintaining ecosystem stability, especially for carnivore top predators such as the common snook and the tarpon.

**Reference Sites Jonuta-Catazaja and Tres Brazos**
Ecological importance and conservation objectives: Very High and High

- Vegetation:
  - Riparian tree *Salix humboldtiana*, tropical swamps tree *Hamaetoxylum campechianum*, palm *Sabal mexicana*, free-floating aquatic plant *Pistia stratiotes* and liana *Dalbergia tabascana*.
  - Tres Brazos. Mangrove trees *Laguncularia racemosa*, *Avicennia germinans*, *Rhizophora mangle*, and *Conocarpus erectus* (protected species by Mexican and international lists), tropical swamps trees *Bucida buceras*, *Pachira aquatica*, *Annona glabra*, and freshwater marshes.
- Fauna: Manatee (*Trichechus manatus*), crocodile, sea otter, common snook, and tarpon for both sites, and Jack Dempsey cichlid (*Rocio octofasciata*) and pale catfish (*Rhamdia guatemalensis*) for Tres Brazos.
- Processes: Primary productivity, connectivity, and sediment transport.

Rationale: This area exhibits species that require regular flooding periods at different magnitudes such as the monodominant forests of *Haematoxylum campechianum* and *Pithecellobium lanceolatum*, or the mangroves species and freshwater marshes. Fish communities have a high composition and richness, many of these are endemic and of economic importance, and represent 19% and 39% of the total fish richness in the area. This region harbors two important Ramsar sites: The Catazaja Lagoon System and *Pantanos de Centla* (Centla Swamps), the last also being a Biosphere Reserve. Both of them are refuge and breeding areas for waterfowls.

Expert recommendation: The conservation of ordinary low flows at 90% and 85% of the mean monthly flows during dry and wet seasons, respectively. A reduction of the river flow above 10% during the dry season would compromise the integrity of the palm *Acoelorraphe wrightii* community that requires wet conditions for seed dispersal. During the wet season, a reduction of the flows above 15% would compromise the high pulse and flood timing that would decrease the surface of the seasonal wetland, losing blue carbon and promoting a greater exposure to saline intrusion. A reduction above 20% would limit the distribution area of the manatee populations due to a reduction of the minimum depth necessary for its movement along the river and the adjacent seasonal wetlands in its floodplain (longitudinal and lateral connectivity) between the lower and middle basin. The extensive flooding areas in these sites provide a high diversity and heterogeneity of habitats, in high pulse and peak flow events season, connecting habitats for most aquatic species as feeding, growing, and protection against predation.

---

## 4. Discussion

The results here presented reveal clear signs of quality on eflows implementation at an administrative level. On the one hand, the metric of comparison of the EWR established against the natural parametrized frequency of occurrence of hydrological conditions turned out to have more meaningfulness to long-term variability than the MAR. This outcome was expected because the reserves were integrated based on the occurrence of the eflow compo-

nents that included the extremes of both the low flows and floods [20–22]. Furthermore, although in relative terms some of the established EWRs exceeded the natural parametrized occurrence, this is because those reserves were set in volumes beyond the theoretical recommendation based on the eflows Mexican Norm implementation. This outcome seems to be related to the commitments of water downstream either by productive uses (i.e., Embalse Zimapán unit with a hydropower dam in Panuco hydrological region) or the environment due to the presence of protected areas or wetlands of international importance (i.e., one basin –San Nicolás B– in Costa de Jalisco and 28 basins from Grijalva-Usumacinta) [69].

On the other hand, the protection enacted to the flow regime by the EWRs could be described and grouped according to different levels of quality. First, those basins where the difference between the EWRs established and the theoretical recommendations are considered as marginal (≥90% equality); 169 out of the 278 basins (61%) were found in this class. On a second level, there is a group of 20 basins whose EWRs met theoretical volumes at a good level (≥80%). The third group encompassed 13 basins where theoretical volumes were met at an acceptable level (≥70%). A total of 202 EWRs met at least this level (73%). Likewise, these levels were found in the residuals' examination between the theoretical volumes and the EWRs established, where 179 of the basins exhibited ≤5%, 207 had ≤10% and 228 showed ≤15%; in other words, 82% of the EWRs had residuals lesser than these thresholds.

Although the general level of fulfillment (EWR meeting) reached by the NWRP depicts the same level of commitment stated in the Mexican policy programs [17–19], there are certainly differences between the theoretical and the established volumes. Such differences detected could have been given due to either a difference in environmental objectives or management classes (nationwide vs. on-site assessed) or to the "real" eflow method outcomes (desktop ecohydrology-based vs. expert panel) [20,21]. While in the present study the quality assessment of the EWRs was conducted based on a desktop method as a standard eflow determination, in some cases, the final EWR established followed the recommendation of an expert panel in the context of holistic assessments [27,28]. That is the case in some of the basins assessed in Costa de Jalisco, Panuco, Papaloapan, and Grijalva-Usumacinta (e.g., Box 3) [20,21].

**Box 3.** Environmental flow implications on Catazaja Lagoon System connectivity dynamics.

Historically, the Catazaja Lagoon System was largely a seasonal wetland that used to dry up entirely during the dry season (November to May) [65]. Dependent on the Usumacinta's high flows, pulses, and peak flow events during the wet season (~2000 $m^3$/s or greater), water and sediments entered the system through multiple branches from the main stem to support these seasonal flooding dynamics [65]. However, a series of levees were built in the early 1990s for keeping the wetland permanently flooded and thus promoting fisheries and tourist activities year-round (Figure 9) [65]. To understand the implications of the eflow recommendations on the wetland flooding dynamics, connectivity was assessed through hydraulic modeling between the Usumacinta's main stem and adjacent wetland system [70]. Furthermore, the river–aquifer interaction (vertical connectivity) was monitored and assessed in 10 shallow wells and one piezometer during the dry and wet seasons [65].

A two-dimensional hydraulic model was developed by using the open-source Iber (https://iberaula.es/space/54/downloads). First, one of Usumacinta's feeding branches to the Great Lagoon of Catazaja (*Laguna Grande de Catazajá*) representative of the seasonal connectivity dynamic was identified. Second, topobathymetry, water, and suspended sediments sampling were taken in the Usumacinta's main stem, and in the branch confluence, to model the river's shape, depth, and hydraulic capacity associated with the flows and water levels that connect to the lagoon. The bathymetric survey was conducted at a scale of 1:2000; 3168 points were sampled in 19 ha (tracks every 20 m and 1 m data record frequency) with a GPS SmaRTK for global positioning, an echo-sounder Hi-Target HD-380 for channel depth, and a multiparametric sounder Hanna HI-9828 for data calibrating. Third, a flooding model was built grounded on digital elevations previously generated based on LiDAR open data administrated by the Mexican National Institute of Statistics and Geography (INEGI). River depth obtained by the on-site topobathymetry and flows from gauging station "Boca del Cerro" (code 30019) were used to calibrate and estimate the all-the-branches depth and the whole wetland system flooding model.

Catazaja Lagoon flooded surface was derived from the Usumacinta's main stem flows at 1500, 2000, 3000, 4000, 5000, 6000, 7000, and 9000 $m^3$/s recorded by the gauging station. Finally, the main channel depth and the flooded surface for the intermediate flow rates (hydrology-based eflow requirements, Mexican Norm's Appendix D application 2 [22]) were obtained based on the equation interpolation ($Y = 3E - 11x^3 - 3E - 7x^2 + 0.0014x + 7.55$, $R^2 = 0.99$; $Y = 576.6 \ln(x) - 3974.2$, $R^2 = 0.97$) (Table 1). The hydraulic parameters outcomes were evaluated and discussed by the expert panel in the context of the Mexican Norm's Appendix F (holistic method, Box 4).

**Box 3.** *Cont.*

In terms of connectivity and based on flow records from 1949–2008 and 2010–2014, the river presented a mean annual base flow of ~620 m$^3$/s and 32% BFI [44,46]. Even in the driest month ever recorded, the river has not stopped flowing (~300 m$^3$/s in May 1963). Likewise, from 650 to ~2000 m$^3$/s water maintains flowing within the main stem and longitudinal connectivity is guaranteed. According to our flooding model, low flows during the wet season for dry and average conditions ranged from 364–484 km$^2$. However, at a lateral connectivity level, the levee on the branch impedes an exchange of water and sediments between the river and the Great Lagoon of Catazaja until a rate of ~3000 m$^3$/s is surpassed. To guarantee the full lateral connection between the river and the wetland, a set of peak flow events based on greater magnitudes is required. Grounded on our model, seasonal flooding dynamics are secured for 729–1142 km$^2$ from high flow pulse to large floods (1- to 10-year return period; medium-size flood at a 5-year return period is the hydrologic parameter to delimitate the river's legal space or public domain in Mexico). Likewise, the shallow wells and piezometers samples and results showed that there is on average 1 m of rising groundwater. This finding revealed the contribution of groundwater on the flooding dynamics, and therefore, the Catazaja Lagoon System dependency on vertical connectivity. Altogether, eflow implementation guarantees the timing of the flows to sustaining the river connectivity condition from high conservation status to free-flowing in 99% of its network (~7130 km) [8,69,71].

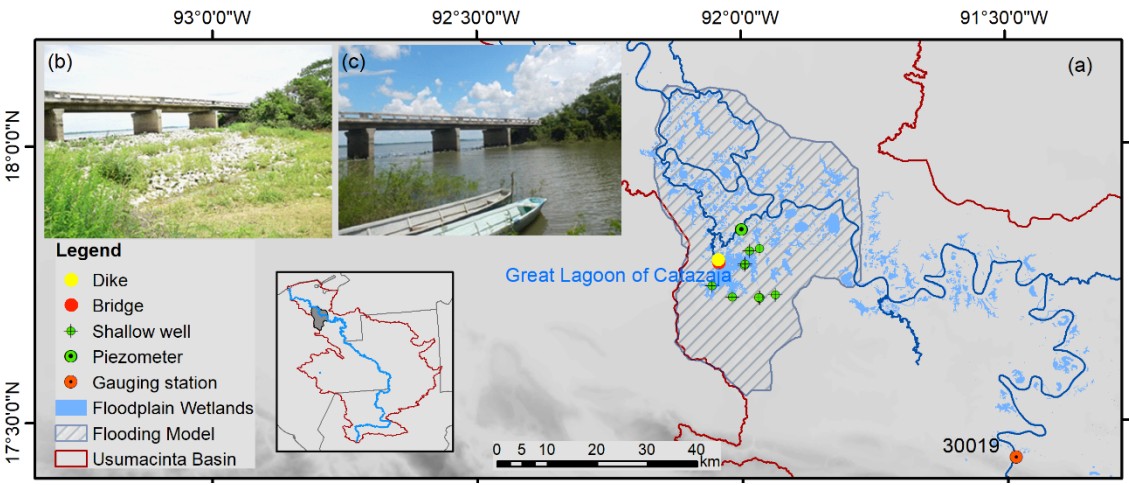

**Figure 9.** Location of the Great Lagoon of Catazaja (**a**). Levee example between one of the Usumacinta river branches and the lagoon, and its effect for keeping the wetland permanently flooded for both dry (**b**) and wet (**c**) seasons.

**Table 1.** Environmental flow components, parameters, and metrics for the connectivity of Usumacinta's main stem (gauging station code 30019) with the Great Lagoon of Catazaja.

| Environmental Flow Component | Parameter | Usumacinta Main Stem | | Catazaja Lagoon |
|---|---|---|---|---|
| | | Discharge (m$^3$/s) | Mean Depth (m) | Flooded Surface (km$^2$) |
| Low flows for dry condition | Dry season | 650 | 8.4 | - |
| | Wet season | 1851 | 9.3 | 364 |
| Low flows for average condition | Dry season | 803 | 8.5 | - |
| | Wet season | 2279 | 9.6 | 484 |
| High pulse | 1.0-year return period | 3488 | 10.1 | 729 |
| Bankfull | 1.5-year return period | 4913 | 10.9 | 927 |
| Medium-size flood | 5.0-year return period | 6409 | 12.6 | 1080 |
| Large flood | 10.0-year return period | 7141 | 13.9 | 1142 |

Concerning the reference values, EWRs were found at a median of 73% MAR for the natural parametrized occurrence of the hydrological conditions, 58% MAR for environmental objective class "A", 41% for a class "B", 33% for a class "C", and 24% for a class "D". In general, this outcome turned out to be consistent with previous experiences in the country, although 3–6% lesser in all the environmental objectives [22]. However, the differences are greater in comparison to the recommended values by other authors, e.g., the Montana method [72] or the Environmental Flow Duration Curve [73,74], which recently have been used to assist countries to estimate the environmental water needs

and incorporate it into the "water stress" indicator 6.4.2. of the Sustainable Development Goals [47]. Although in such cases the values were quite similar for an environmental objective class "A" (~60%), for the rest of the classes the Mexican values were higher, and the differences ranged 13–16%. This is explained by the frequency-of-occurrence of eflow components approach and the parametrized thresholds, which were developed to manage the Mexican hydrological conditions and particular implementation challenges [22]. Unlike other methods, the components of low flows and peak flow events are adjusted not to a proportion of their magnitude but their managed occurrence [22]. In this sense, this novel approach depicts better short- and long-term variability of flows even at low environmental objectives classes, which was confirmed by on-site holistic studies reported in the literature [21,22,31].

About the reference values of the supporting indices of flow variability, in general, these were similar to the previously reported [21,22,31]. The exception was found for the median +25% parameter (third quantile) where according to the literature for the country the CV was ~280%, the BFI ~22%, and the CVB ~120% [21,22,31]. These differences seem to be related to outliers; if they are removed the values from the literature adjust to the ranges of 107–240% CV, 3.5–23% BFI, and 4–80% CVB, which are fairly close to those obtained for this research. As a point of further comparison, these outcomes are similar to the reported for the South African rivers (i.e., CVB = 6–35%, median 12%) [44].

### 4.1. Implications and Limitations

Given that flow modification and overexploitation of water resources have a direct effect on the dramatic trends of the freshwater biodiversity loss [3,7,10], eflows implementation has been pointed out as an urgent measure [10,14,75]. The Mexican NWRP came out as a public policy commitment to enact water for ecological protection before its use goes beyond sustainable limits of abstraction [20,22,31]. This program has been recognized by eflows science and as a state-of-the-art practice [10,26–28]. The results provided in this research add to the existing literature on the eflow assessment outcomes of the country. Here, a baseline of EWRs is provided to evaluate both the future path of the NWRP but also their implementation on the ground. Altogether, these results contribute to the foreseen adaptive management from both the Mexican eflows standard and the current network of EWRs.

The EWR and flow variability indices reference values obtained in this research contribute to the knowledge of the Mexican rivers' ecohydrology [21,22,31]. Along with the existing literature, these outcomes suggest the revision on the suitability of recommending more detailed values for the "look-up-tables" of both the characteristic percentages of EWR as well as to the frequency-of-occurrence management factors, per environmental objective class, stated in the Mexican eflows' Norm (Appendix D, applications 1 and 2) [21,22,31]. Unlike perennial rivers, flow cessation is a key feature of intermittent rivers and ephemeral streams (IRES) that makes them hotspots of biodiversity because of the complex mosaic of flowing/nonflowing water and terrestrial habitats for the support of distinctive aquatic, semiaquatic, and terrestrial species [76–83]. Concerning this issue, it is already known that the greater the variance between dry and wet seasons the greater the wet conditions dependency between streamflow types (ephemeral streams > intermittent > perennial rivers), and this tendency is reflected in the EWR relative volumes (perennial > intermittent > ephemeral) [31]. The reference values of the Mexican standard need to be adjusted to reflect such characteristics.

One clear benefit about the EWRs is the potential that they have in the long run to protect freshwater ecosystems that depend on the flow regime integrity while ensuring sustainable limits of water abstraction for people and economic activities at the river basin level. To date, the existing network of EWRs contributes with 169,900+ hm$^3$ of surface water to incorporate eflows into the Sustainable Development Goal's "water stress" indicator 6.4.2 (~65% of the total reported) [42,47,84], which drain throughout ~25% of the Mexican territory (484,000+ km$^2$) [69]. This means water security for ~45 million



people and flows connectivity for 82 protected areas (175,000+ km$^2$) and 64 wetlands of international importance (Ramsar sites; 47,000+ km$^2$) for up to 50 years [6,32–41] [85,86]. Based on detailed eflow assessments in 25 sites from 2012–2015, water needs from 90+ freshwater-dependent species (40+ under protection) were directly considered [21], and more recently it was estimated that the whole initiative holds the potential for protecting or improving the conservation status of over 450 freshwater fish species [87].

Furthermore, based on the latest world's free-flowing rivers (FFR) assessment [8,71], these EWRs' network mean the legal protection to the flow regime, that is to say, the temporal or fourth dimension of connectivity [88] in ~44,000 km (~33% of the Mexican river network assessed), and ~52,000 km more that do not have yet any figure of flow protection [69]. These drainage river basins represent a strategic opportunity to continue with the policy of enacting water for ecological protection, as stated in the Mexican National Water Plan 2020–2024 by the current government administration whose goal is to increase the EWR network from 295 to 448 [43]. For some of them, eflow technical studies are already developed, e.g., Cerrada Laguna Salada-El Borrego or the Todos Santos-Los Barriles Corridor in the peninsula of Baja California, Acaponeta-Piaxtla between Sinaloa and Nayarit, and Copalita-Zimatán-Huatulco in Oaxaca [20,21].

Another implication of setting these strategic environmental allocations consists of limiting future unsustainable water abstraction and infrastructure [15,22]. To date, EWRs have caused that two hydropower projects were denied by the Mexican government [32–41,89]. The first was in 2014 for "Las Cruces", a hydropower project promoted by the Federal Electricity Commission that aimed to be placed at the San Pedro Mezquital river (Nayarit) [90]. In this case, detailed studies of hydrology, sediment transport, and economics associated with the natural flow regime demonstrated that the operation of the dam would not be consistent with the EWR [24,25,27]. Similarly, the second case was in 2018 where the "Santo Domingo" hydropower project, promoted by a private company, was rejected due to pretending to be placed at the Santo Domingo and Lacantun rivers (Chiapas), tributaries of the Usumacinta, and contradicted the EWR decree and underlying Environmental Flow Assessment Study [65,91,92] (Box 4). In both cases, the EWR and the underlying eflow studies have proved to be efficient administrative tools for setting the rules of the basin water usage sustainably, and for ensuring the rivers' temporal connectivity protection.

Among the limitations, first, it should be emphasized that the rainfall-runoff models (indirect method in 211 basins) were used to overcome the lack of gauged flow records. Although this did not impede the application of the eflows' Norm (Appendices A and B), it brought uncertainty to the EWR volumes and flow variability indices for delivering reference values consistently per stream type [93]. Furthermore, the EWR volumes coming from this implementation type of the eflows' Norm do not consider the floods' proportional part, thus would explain that the results here presented turned out to be 3–6% lesser for all the environmental objectives based on previous experiences in the country [22]. One option for further research and to overcome this challenge would be to gather the eflow and flow variability outcomes from this research with those from other efforts, investigate per-stream flow type its hydrological condition dependency, and deliver reference values with statistical robustness [31].

Second, at a broader level, eflows enacted protection as an administrative measure alone, within the water allocation system, is not enough to manage and secure the spatial dimensions of connectivity: Longitudinal along the river (upstream-downstream linkages), lateral between this and its floodplains, and vertical e.g., surface water and groundwater interactions [53,88,94,95]. Given that the Mexican water allocation system's norms and policy are annual based [62,63], the main challenge identified in EWR implementation consists of managing the volume into higher temporal resolution. Certainly, since the system's eco-hydrological functioning and integrity depends on the interaction through time (the fourth dimension) of the spatial longitudinal, lateral, and vertical connectivity, the legal protection of the temporal connectivity secures this multidimensional interaction [53,88,94,95]. Even

though the eflows' Mexican Standard and on-site assessment studies outcomes provide regime recommendations, water titles are provided at an annual scale.

In this regard, complementary policies and regulations are needed, in particular those that link water and territory. In this sense, the EWR decrees mandate the integration of the eflow recommendations into protected areas management programs e.g., the San Pedro Mezquital reserve where clear species and habitat eflows-related objectives and requirements are indicated [22,89]. Likewise, in 2019 was published a new decree mandating the integration of the available water after enacting the EWRs into the national water planning, as well as to set up the operative rules for ensuring the EWRs fulfillment in terms of quantity, quality, and timing [96]. This executive order means to foresee environmental water needs through water management plans at an on-site scale to set up operative rules, and governance actions to monitor and evaluate the performance of the EWR objectives from the river basin councils. A goal in this regard has been stated by the current government administration in the National Water Program 2020–2024 [43]. In the short to the mid-term, this implies the need of testing the performance of the EWRs against the flow–ecology relationships and the societal use of water that grounded the eflow assessments, and an opportunity for adjusting the environmental objectives and/or eflow recommendations (i.e., adaptive management). Furthermore, whether for protected areas management programs or water management plans, the integration of specific eflow objectives linking species, habitat requirements including connectivity, and societal use of water goals at a proper time-scale is an opportunity to overcome the current EWR administrative limitations (e.g., Box 4).

**Box 4.** Flow–ecology relationships between the Usumacinta river and the adjacent wetlands.

Eflow implementation guarantees a diverse set of ecological processes that are essential to maintain the aquatic ecosystems' productivity and species biodiversity of the basin. Consequently, the expert panel formulated the following flow–ecology relationships (Figure 10) based on the eflows' recommendation and their implications in terms of connectivity previously provided (Box 3, Table 1) [65,70]. During dry season low flows (~600–800 $m^3$/s), biogeochemical oxidation processes occur in wetland soils, allowing the reincorporation of nutrients and decomposition of organic matter over larger areas. Root development and nutrient uptake by plants take place, while floodplain tree species begin to flourish and produce seeds. It is also worth mentioning the synchronization of productive activities such as yearly crop planting occurs when water levels drop allowing the land to be plowed for ~6–8 months.

At flows between ~2000 and 3000 $m^3$/s, seedling germination, dispersal and establishment take place, while floodplains of tropical swamps tree *Haematoxylum campechianum* regenerate triggered by the yearly wet season-low flows. These water volumes allow the redistribution of sediments as well as processes for seeding dispersal of free-floating plants (*Pistia stratiotes*) and migration of economic fish species such as the common snook and tarpon. Concerning economic activities, cattle producers and farmers move their cows and crops before flooding processes occur showing synchronization of economic activities with the natural flow regime. At a discharge of ~4000 $m^3$/s, ~10.4 m of mean depth are achieved, water and sediments surpass by 1.6 m the levee on the branch, and the Great Lagoon of Catazaja is fully connected. This water level guarantees a yearly exchange of manatees, common snook, and tarpon between the river and the wetland. At these flow rates, seed dispersal of tropical swamps trees *Haematoxylum campechianum* and *Pithecellobium lanceolatum* occurs while flooding allows weeds to be controlled. In addition, the revitalization of oak communities occurs, as empirically observed, require yearly regular flooding (1- to 1.5-year return period). Other processes are promoted, such as the release of nutrients from the soil, their transport via floodplain, and the deposition of sediments. Sports-recreational activities such as the tarpon tournament and bird watching in nesting areas are also carried out.

Finally, at a peak flow event of ~6000 $m^3$/s or greater, full lateral connectivity occurs. At this level, low-frequency temporal wetland ecosystems offer new feeding and shelter habitat for the recruitment of populations of macroinvertebrates and fish. This period of maximum flooding is key for the resurgence of extensive areas of permanent, seasonal, and temporary freshwater wetlands that characterized the Catazaja lagoon system.

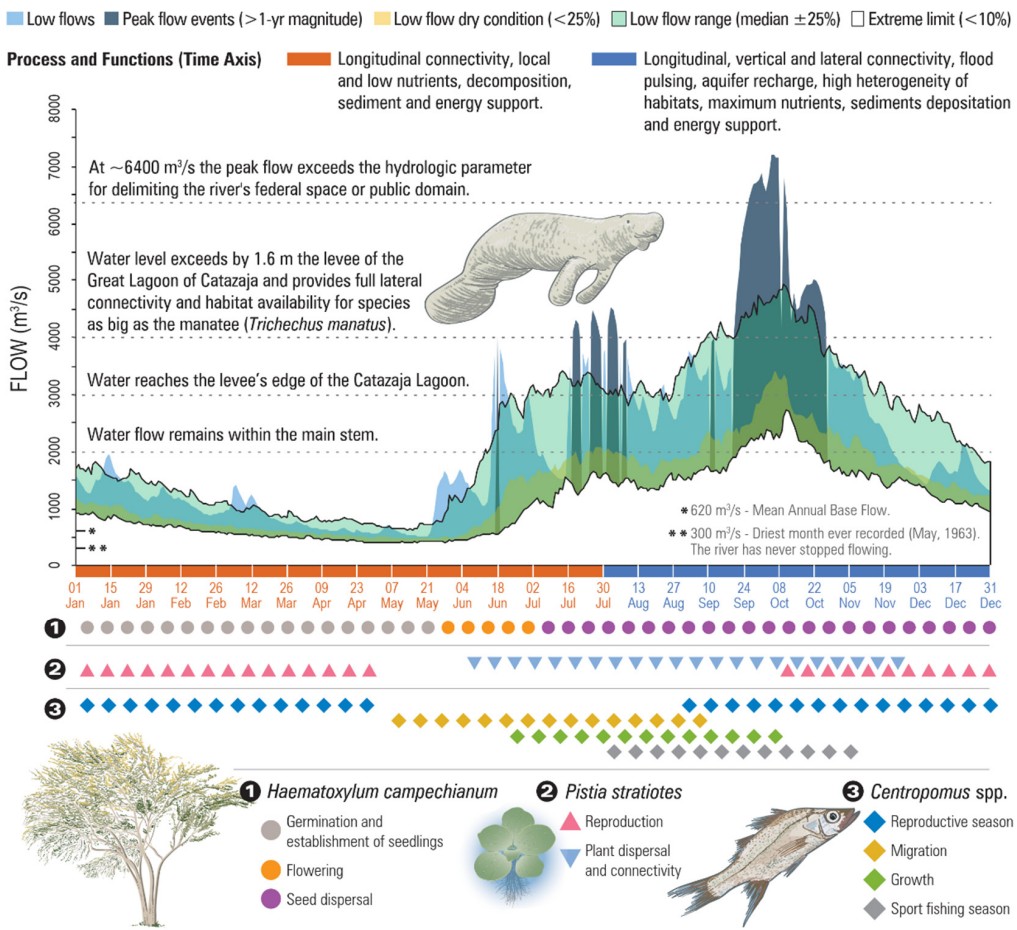

**Figure 10.** Conceptual hydrograph of environmental flow components and flow-ecology relationships with targeted conservation objectives. For more detail, refer to the Usumacinta's hydraulic modeling, sediment transport, and environmental flow assessment reports provided as Supplementary Material (only in Spanish).

### 4.2. Outlook and Recommendations

Since the eflow studies that underlie the EWR policy were conducted through a strategic hierarchical assessment approach, the level of certainty is differentiated between holistic ecohydrological desktop or expert panel [20,21,27,28,30,31]. To guarantee the implementation and adaptive management of the eflows protected, an *Environmental Water Reserves Monitoring Network* (RedMORA) was formed in 2018 within the National Association of Universities and Higher Education Institutions to design and operate a nationwide research-driven system to assess the performance of the reserves enacted [97]. To date, this network is integrated by 65 academics, water and ecosystem managers from universities, research centers, non-governmental organizations, CONAGUA and CONANP staff, and it is organized by groups at three different levels of incidence [98]. The first is local or subnational focused on testing flow-ecology and societal use working hypotheses underlying the reserves in Costa de Jalisco, Papaloapan, Panuco, San Fernando-Soto La Marina, San Pedro Mezquital, and Usumacinta regions [98]. The second level is thematic or cross-sectional between regions oriented to six tasks to manage the knowledge generated on-site around the environmental water science and practice: Capacity building and human resources, citizen science and communication, data engineering and transfer, the interaction between government, academy, and society (policy implementation and governance), water availability alternatives, and the monitoring and assessment system itself [98]. The third level is one coordinator committee in charge of the administrative management of the network, and the interactions between the on-site and thematic groups with central authorities (CONAGUA and CONANP) [98]. Outcomes from the RedMORA

are expected to provide evidence to scale up the system from 295 to 448 EWR throughout the country and to set up the operative rules and governance actions around its adaptive management and the decision-making around the performance of the environmental water goal of the National Water Program 2020–2024 [43].

Given the achievements as a water conservation mechanism of Mexico's NWRP, efforts have been made to scale the program in the Latin-American region [28]. Countries such as Bolivia, Colombia, Ecuador, Guatemala, Honduras, and Peru are at the startup phase of the WWF's Water Reserves Regional Initiative, which aims to allocate water for the environment through the implementation of eflows in the region's most important water-producing areas such as the Amazon headwaters, Cerrado-Pantanal and the Mayan Rainforest (https://wwf.panda.org/discover/our_focus/freshwater_practice/freshwater_inititiaves/water_reserves_initiative/).

As it was seen with the experience of Mexico, we emphasize that programs alike be developed in coordination and consensus with national and local water management and territory authorities. Furthermore, the process of implementation must be tailored according to the institutional and legal frameworks present in each country; recommendations in this line have been provided also by Harwood et al. (2017) [25]. Conversations are underway with government partners, and for instance, guidelines for eflow determination have been established in Peru [99] and a National Environmental Flows Initiative has been drafted in the country. Capacity building processes in eflows and EWRs have begun in Bolivia, Guatemala, and soon Honduras, as means to establish their National Environmental Flows Agenda, aiming at incentivizing the development of pilot eflows studies based on fieldwork and expert panel experiences.

## 5. Conclusions

In the last years, the Mexican government set the ambitious commitment in public policies of enacting water for ecological protection based on the strategic identification of river basins for feasible environmental water allocation at the country scale. To date, there is an array of eflow assessment outcomes and literature available, from the desktop (eco)hydrology-based to an expert panel and research-driven methods based on on-site information and detailed models. After almost a decade, the progress achieved surpassed the number of basins with EWR enacted by more than 50%.

At a national level, independently of each hydrological region, downstream water commitments, or environmental merits context, around three-quarters of the EWR established met theoretical volumes at an acceptable level according to the Mexican Standard for determining eflows. Furthermore, EWR and basic flow variability reference values are provided for future eflow assessments and environmental water policy progress, and the implications and limitations as a long-term protection tool for high priority conservation values at the river basin scale stated. Although balancing the scope and certainty level of the outcomes with the decision-making in eflow protection remains to be challenging, the strategic hierarchical approach followed in Mexico has proved to be useful and consistent to achieve progress in a relatively short term at a national scale.

Due to the policy commitment of enacting EWR remains in the present government administration, monitoring and assessing eflows performance on the ground is key not only for adaptive management of such reserves but also for providing feedback towards the EWRs about to come. A new EWR stewardship is emerging, and it can be strengthened and empowered by encompassing academics, organized society, basin inhabitants and authorities, which provides open and joint accountability in environmental water performance assessment and legitimates its adaptive management long-term water security and freshwater ecosystems conservation goals.

**Supplementary Materials:** The following are available online at https://www.mdpi.com/2071-1050/13/3/1240/s1, the database for the current national assessment, the Usumacinta hydraulic modeling and sediment transport, and environmental flow assessment reports (only in Spanish).

**Author Contributions:** Conceptualization, methodology, validation, investigation, writing—original draft preparation, and funding acquisition, S.A.S.-R.; formal analysis, writing—review and editing, S.A.S.-R., E.B.-M., D.I.M., M.Z.N.-L., I.N.-F., R.D.V. and I.D.G.M.; visualization, S.A.S.-R. and I.N.-F.; supervision, S.A.S.-R. and, E.B.-M.; project administration, S.A.S.-R., E.B.-M., M.Z.N.-L. and R.D.V. All authors have read and agreed to the published version of the manuscript.

**Funding:** This research was funded by the Alliance of the World Wildlife Fund and the Gonzalo Río Arronte Foundation I.A.P. (WWF-FGRA), grant number A.297, and the Inter-American Development Bank, grant numbers ATN/OC-12827-ME-1 and ATN/OC-15163-ME. The APC was funded by the WWF-FGRA Alliance.

**Institutional Review Board Statement:** Not applicable.

**Informed Consent Statement:** Not applicable.

**Data Availability Statement:** The data presented in this study are available in www.mdpi.com/2071-1050/13/3/1240.

**Acknowledgments:** We are thankful to the Department of Engineering and Binational Affairs, Deputy Director General's Office for Technical Affairs from the National Water Commission. We also thank Eugenio Barrios Ordóñez, Rafael Sánchez Navarro, Mario López Pérez, Alfredo Ocón Gutiérrez, Horacio Rubio Gutiérrez, Rafael Rosales González, Ricardo Villón Bracamonte, Fabiana Rosales Ángeles, Aarón Schroeder Aguirre (†), Adriana Guerra Gilbert, and Anuar Martínez for their significant contribution to the National Water Reserves for the Environment Program design, funding acquisition and operation, and Rodolfo Pérez Orduña for the graphic design work in the conceptual hydrograph of the Usumacinta river.

**Conflicts of Interest:** The authors declare no conflict of interest. The funders had no role in the design of the study; in the collection, analyses, or interpretation of data; in the writing of the manuscript, or in the decision to publish the results.

## Abbreviations

| | |
|---|---|
| BFI | Baseflow index (%) |
| CONAGUA | National Water Commission |
| CONANP | National Commission of Natural Protected Areas |
| CV | Coefficient of variation (%) |
| CVB | Overall index of variability of flows (%) |
| Eflow | Environmental flow |
| EWR | Environmental water reserve |
| EwrNat/A/B/C/D | Environmental water reserve according to the natural-parametrized frequency of occurrence of the low flows and the peak flow events, or for the environmental objective classes "A", "B", "C", and "D" ($hm^3$/year) |
| FFR | Free-flowing river |
| FrA/B/C/D | Flood regime (1-, 1.5, and 5-year return period peak flow events) for the environmental objective classes "A", "B", "C", and "D" ($hm^3$/year) |
| HR | Hydrological region |
| INEGI | National Institute of Statistics and Geography |
| IRES | Intermittent rivers and ephemeral streams |
| MAR | Mean annual runoff ($hm^3$/year) |
| NatPFoO | Natural-parametrized frequency of occurrence of the low flows and the peak flow events |
| NMX-AA-159-SCFI-2012 | Mexican Norm that establishes the procedure for environmental flow determination in hydrological basins |
| NWRP | National Water Reserves for the Environment Program |
| Q | Flow discharge ($m^3$/s) |
| RedMORA | Environmental Water Reserves Monitoring Network |
| SolfNat/A/B/C/D | Seasonal ordinary low flows for the natural-parametrized frequency of occurrence for very dry, dry, average, and wet conditions, or for the environmental objective classes "A", "B", "C", and "D" ($hm^3$/year) |
| WWF | World Wildlife Fund Inc. |

## Appendix A

EWRs calculated based on daily flow observed records for both the low flow and flood regime eflow components ranged reference values from 65–76% MAR (70% median) for the natural parametrized frequency of occurrence, while for the characteristic volumes per environmental objectives ranged from 47% to 61% for a class "A" (median 52%), 27–44% class "B" (median 33%), 17–36% class "C" (median 24%), and 11–30% class "D" (median 18%; *n* = 69; Figure A1). Flow variability supporting indices reference values ranged from 116% to 234% CV (median 159%) between dry and wet seasons, 8–23% BFI (median 14%), and 5–30% CVB (median 12%) (Figure A2).

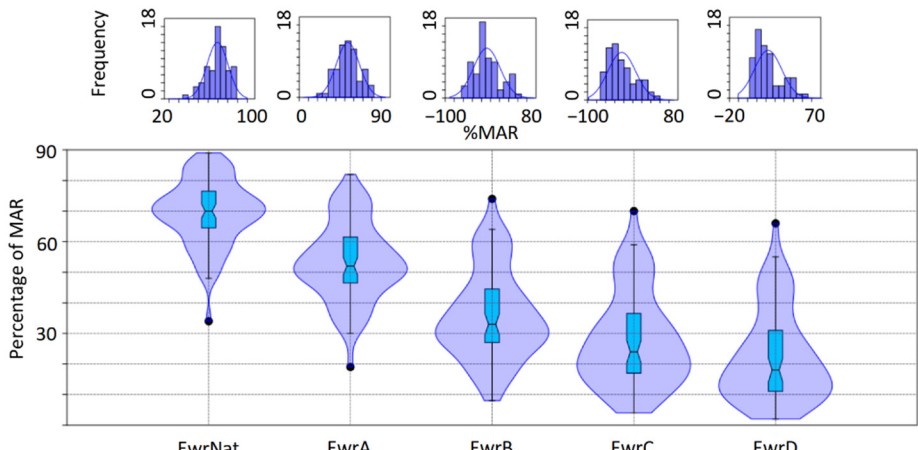

**Figure A1.** Environmental water reserves (EWR) reference values in percentage of mean annual runoff (MAR) based on a central range distribution approach on the environmental flows' Mexican Norm (NMX-AA-159-SCFI-2012) implementation outcomes, according to flow observations from gauging stations (*n* = 69). Values are given for the natural parametrized reference (EwrNat) and the environmental objectives (A−D).

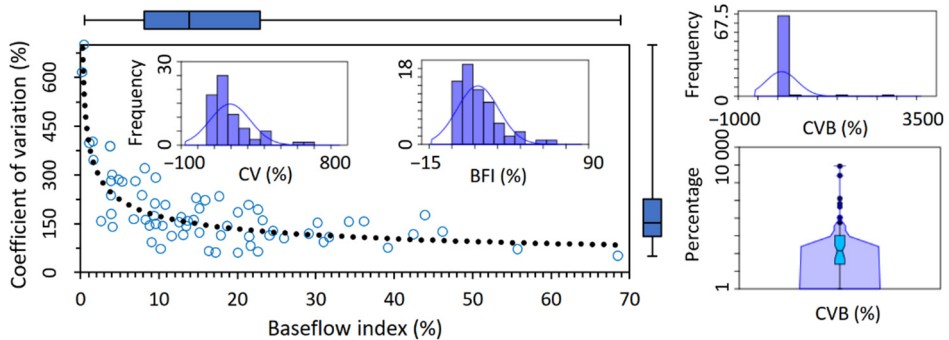

**Figure A2.** Relationship between the coefficient of variation seasons' indicator (CV) and the baseflow index (BFI) (y = 406.76x$^{-0.37}$, R$^2$ = 0.51), the overall index of flow variability (CVB; boxplot displayed at log scale), and their reference values calculated based on a central range distribution approach. Data obtained from flow observations in 69 gauging stations.

Likewise, EWRs reference values from 216 river basins where the method was implemented in monthly-scale rainfall-runoff corresponding models ranged from 68–78% MAR (74% median) for the natural parametrized frequency of occurrence, while for the characteristic volumes per environmental objectives ranged from 51–65% for a class "A" (median 59%), 35–51% class "B" (median 44%), 26–43% class "C" (median 35%), and 19–35% class "D" (median 26%; Figure A3). Flow variability supporting indices reference values

ranged from 108% to 202% CV (median 146%) between dry and wet seasons, 2–15% BFI (median 8%), and 7–98% CVB (median 19%) (Figure A4).

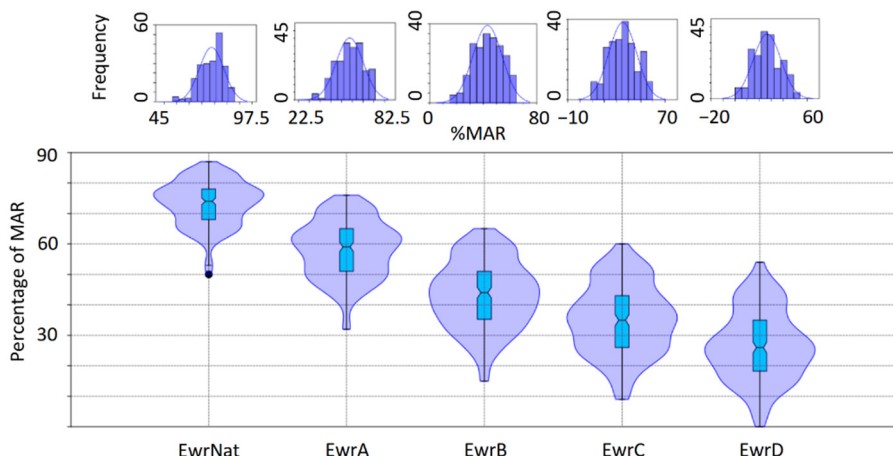

**Figure A3.** Environmental water reserves (EWR) reference values in percentage of mean annual runoff (MAR) based on a central range distribution approach on the environmental flows' Mexican Norm (NMX-AA-159-SCFI-2012) implementation outcomes (rainfall-runoff models, *n* = 216). Values are given for the natural parametrized reference (EwrNat) and the environmental objectives (A−D).

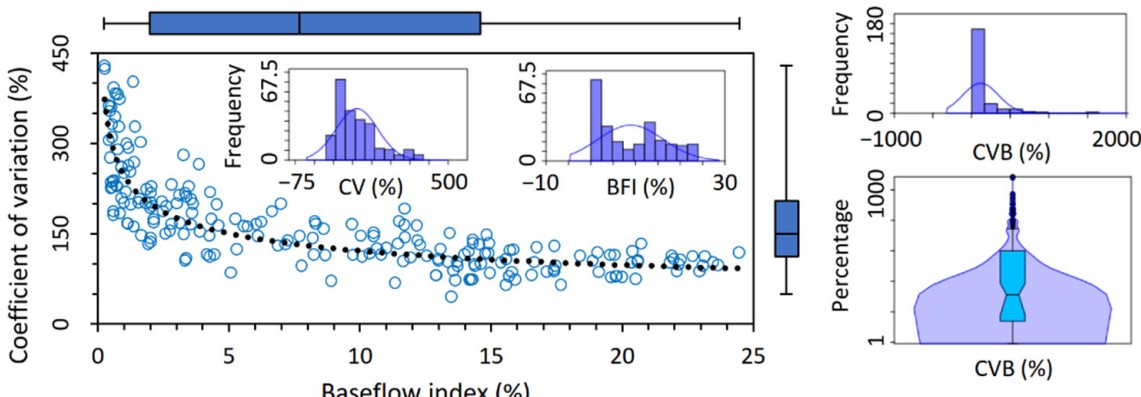

**Figure A4.** Relationship between the coefficient of variation seasons' indicator (CV) and the baseflow index (BFI) (y = 246.83x$^{-0.31}$, R$^2$ = 0.72), the overall index of flow variability (CVB; boxplot displayed at log scale), and their reference values calculated based on a central range distribution approach (rainfall-runoff models, *n* = 211).

## Appendix B

According to daily flow observed records from 69 gauging stations, for both the low flows and flood regime eflow components, the seasonal ordinary low flows corresponding volumes ranged from 49–67% MAR (median 61%) for the natural parametrized frequency of occurrence, 32–52% (median 43%) for an environmental objective class "A", 17–38% (median 26%) for "B", 11–33% for "C" (median 21%), and 6–28% for "D" (median 16%) (Figure A5). Likewise, flood regime reference values corresponding volumes ranged from 8–12% MAR (median 10%) for an environmental objective class "A" (natural occurrence), 5–8% (median 7%) for "B", 3–5% (median 4) for "C", and 2–3% for "D" (median 3%) (Figure A6).

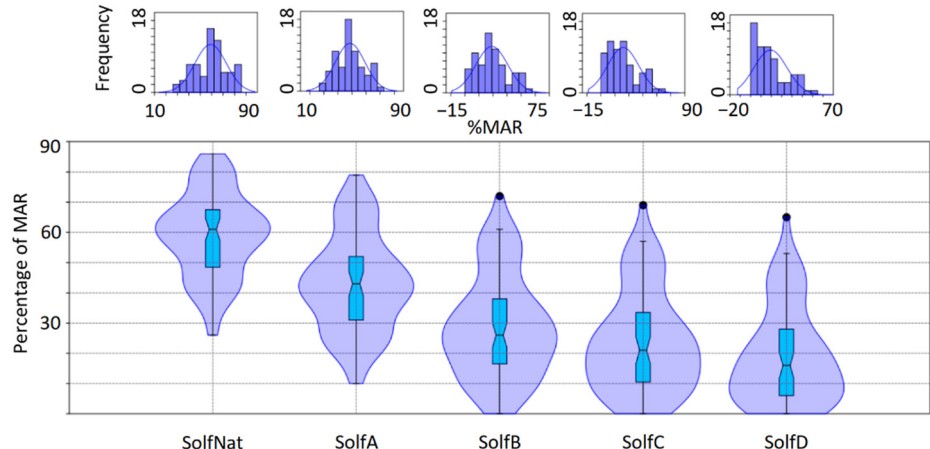

**Figure A5.** Seasonal ordinary low flows (Solf) reference values in percentage of mean annual runoff (MAR) based on a central range distribution approach on the environmental flows' Mexican Norm (NMX-AA-159-SCFI-2012) implementation outcomes according to flow observations from gauging stations (*n* = 69). Values are given for the natural parametrized reference (SolfNat) and the environmental objectives (A−D).

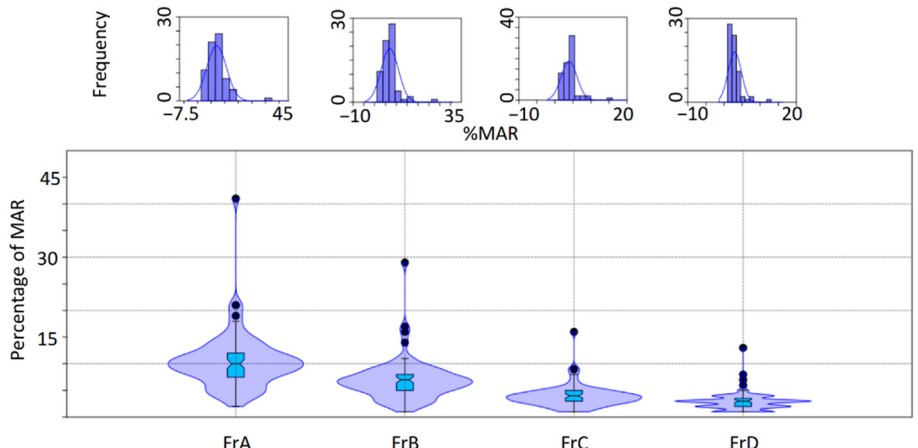

**Figure A6.** Flood regime (Fr) reference values in percentage of mean annual runoff (MAR) based on a central range distribution approach on the environmental flows' Mexican Norm (NMX-AA-159-SCFI-2012) implementation outcomes according to flow observations from gauging stations (*n* = 69). Values are given for each environmental objective (A−D).

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
