# Peer review of "What Do Environmental Flows Mean for Long-term Freshwater Ecosystems’ Protection? Assessment of the Mexican Water Reserves for the Environment Program"

_sustainability, doi:10.3390/su13031240_

Round 1

Reviewer 1 Report

Dear Authors,

I am writing this to submit my comments on your research article with the following details.

Manuscript title: What do environmental flows mean for long-term freshwater ecosystems’ protection? Assessment of the Mexican Water Reserves for the Environment Program

Manuscript Number: sustainability-1066121

Journal Submitted: Sustainability

General Comments:

I would simply say, that it is a nicely presented study and needs minor revisions. Excellent. The only problem with the manuscript is its abstract and objectives parts. So please take care of those and looks like you are good to go.

Specific Comments:

Title:

Fine and represents the study.

Abstract:

Please enrich your abstract with more findings and furnish what you concluded. The abstract must be revised.

Keywords are not actually the keywords but long terminologies. I suggest you to revise most of them.

Introduction:

The introduction is fine except its ending and the objectives. You should avoid mentioning what you have achieved or what you are presenting here. Rather follow the standard style of writing your objectives.

Materials and Methods:

Methods are done sophisticatedly and provide all the details.

Results and Discussion:

No changes suggested. It is mostly well-written.

Figures and Tables:

All figures and tables are necessary and add value to the manuscript.

Conclusions

Conclusions are to-the-point.

References:

Okay.

Author Response

Esteemed Reviewer # 1,

Many thanks for your feedback. We revised both the Abstract and the objectives in the Introduction section as was suggested. Please refer to the attached "Authors' response" file where we detailed all the amendments made (R.-). We hope you find these issues sufficiently addressed.

Also, be aware that in-line numbers cited in our response letter referred to our revised manuscript. Finally, we would like to stress your attention to note that there is a new figure (#1) with a flow chart summarizing the whole methodological approach, and all figures were revised for increasing the text size where needed, among other things suggested or recommended by Reviewer #2. You will identify all the amendments made in the manuscript easily since the track changes function was used.

We hope you find this revision of our manuscript as satisfying as we find it.

Sincerely,

(on behalf of the authors)

Lead and corresponding author

Reviewer 2 Report

The manuscript is interesting. Nevertheless, it needs some further improvement. In general, there are still some occasional grammar errors throughout the manuscript, especially the article ‘’the’’, ‘’a’’ and ‘’an’’ is missing in many places; please make a spellchecking in addition to these minor issues. The reviewer has listed some specific comments that might help the authors further enhance the manuscript's quality.

  1. Specific Comments

Please include a list of acronyms and abbreviations.

  • Introduction
  • The objectives are not explicitly stated.
  • The authors need to enrich further the background; please highlight the importance of reaching suitable trade-offs among renewable energy sources development in achieving sustainability goals and ecosystem conservation through better management of e-flows. The following literature might be useful in this regard << Environmental Flows Assessment in Nepal: The Case of Kaligandaki River>> and <<Water-energy-ecosystem nexus: Balancing competing interests at a run-of-river hydropower plant coupling a hydrologic–ecohydraulic approach>> you may review other additional relevant references as well.
  • What is the novelty of this work?

  • Methods
  • I would suggest showing the methodological approach through a flowchart
  • Methodology limitations should be mentioned.
  • All variables should be explained.
  • What was the data resolution used in the computations?

  • Results
  • This section is well written.
  • Please increase a bit the text size for all figures.
  • What is the reason for using many text boxes?

  • Discussion

The discussion should summarize the main finding(s) of the manuscript in the context of the broader scientific literature and address any limitations of the study or results that conflict with other published work.

Author Response

Esteemed Reviewer # 2,

Many thanks for your feedback. We improved the manuscript based on your suggestions and recommendations and revised the manuscript thoroughly to amend occasional grammar errors. Please refer to the attached "Authors' response" file where we detailed all the amendments made (R.-). We hope you find all the issues sufficiently addressed or clarified.

Also, be aware that in-line numbers cited in our response letter referred to our revised manuscript. Finally, we would like to stress your attention to note that there is a new figure (#1) with a flow chart summarizing the whole methodological approach, and all figures were revised for increasing the text size where needed, among other things suggested or recommended by you. You will identify all the amendments made in the manuscript easily since the track changes function was used.

We hope you find this revision of our manuscript as satisfying as we find it.

Sincerely,

(on behalf of the authors)

Lead and corresponding author
